# Thoracic Diseases: Technique and Applications of Dual-Energy CT

**DOI:** 10.3390/diagnostics13142440

**Published:** 2023-07-21

**Authors:** Armando Perrella, Giulio Bagnacci, Nunzia Di Meglio, Vito Di Martino, Maria Antonietta Mazzei

**Affiliations:** Unit of Diagnostic Imaging, Department of Medical, Surgical and Neuro Sciences and of Radiological Sciences, University of Siena, Azienda Ospedaliero-Universitaria Senese, 53100 Siena, Italy; giuliobagnacci@gmail.com (G.B.); dimeglionunzia@gmail.com (N.D.M.); dimartino7@student.unisi.it (V.D.M.); mariaantonietta.mazzei@unisi.it (M.A.M.)

**Keywords:** dual-energy CT, breast cancer, pulmonary embolism, acute aortic syndromes, lymph nodes, pleural carcinomatosis, thymoma, lung cancer, ILD, esophageal cancer

## Abstract

Dual-energy computed tomography (DECT) is one of the most promising technological innovations made in the field of imaging in recent years. Thanks to its ability to provide quantitative and reproducible data, and to improve radiologists’ confidence, especially in the less experienced, its applications are increasing in number and variety. In thoracic diseases, DECT is able to provide well-known benefits, although many recent articles have sought to investigate new perspectives. This narrative review aims to provide the reader with an overview of the applications and advantages of DECT in thoracic diseases, focusing on the most recent innovations. The research process was conducted on the databases of Pubmed and Cochrane. The article is organized according to the anatomical district: the review will focus on pleural, lung parenchymal, breast, mediastinal, lymph nodes, vascular and skeletal applications of DECT. In conclusion, considering the new potential applications and the evidence reported in the latest papers, DECT is progressively entering the daily practice of radiologists, and by reading this simple narrative review, every radiologist will know the state of the art of DECT in thoracic diseases.

## 1. Introduction

Dual-energy computed tomography (DECT) is one of the most promising technical innovations in the field of computed tomography (CT). It represents an emerging quantitative and functional technique that may provide considerable advantages, by overcoming some of the limitations of conventional contrast-enhanced CT. DECT extends the capabilities of conventional CT beyond the established densitometric analyses; it was first hypothesised in 1973 by Hounsfield GN and refers to the simultaneous/near-simultaneous acquisition of CT attenuation data derived from two different ranges of X-ray energy levels, avoiding misregistration artefacts between the two imaging sets [1,2,3,4,5,6,7,8,9,10]. There are several technologies behind DECT nowadays. Regardless of the technical approach to obtaining DECT data (single-source twin-beam, single-source rapid-switching, single-source sequential, dual-layer, and dual-source), data post-processing allows the characterization of a specific material, the elaboration of virtual monoenergetic images (VMI) at a selected energy level, and the extrapolation of material-selective images, namely, material decomposition (MD) images, including iodine map, virtual no calcium and virtual non-contrast (VNC). DECT offers both some diagnostic advantages and a reduction in radiation dose and in the volume of contrast material. By using a low-energy VMI (between 40 and 55 keV), it is possible to achieve a significant increase in iodine attenuation, resulting in better image quality, even with a large decrease in the volume of injected iodinated contrast media, ultimately reducing nephrotoxicity (especially in cancer patients exposed to nephrotoxic drugs or previous nephrectomies). Moreover, it is well documented in the literature that DECT of the chest is feasible without additional doses and any significant difference in image noise. Moreover, the use of VNC imaging can even lead to a significant reduction in the total applied dose, since pre-contrast scans become no longer necessary [11,12,13,14,15]. Another possible benefit comes from using the high-energy VMI to reduce beam hardening artefacts, which can result in the thorax from the presence of dorsal spine stabilization in scoliosis [16,17,18].

Obviously, it must be kept in mind that DECT, like all radiological techniques, has its own limitations that must be known in order to make the best use of this diagnostic method [19].

The applications of DECT are widely described in the literature, and the evidence of its benefits is now well known [20,21,22,23,24,25,26,27,28,29,30,31,32,33,34,35,36,37,38]. In oncology, this technique can increase the conspicuity of hypervascular or hypovascular lesions in the context of the surrounding parenchyma with an acceptable signal-to-noise index, by using low VMI (close to the K-edge of iodine, 33.2 keV). Quantitative analyses of such images allow the radiologist to objectively evaluate iodine uptake in a target lesion during the characterization phase or the assessment of response to treatments. Effectively, DECT has particular relevance for evaluating response to the treatment of highly perfused tumors [39,40,41]. Non-oncologic applications in thoracic diseases include the assessment of interstitial lung disease, even if evidence from the literature is scarce, and cardio-vascular applications (especially pulmonary hypertension, acute or chronic pulmonary embolism, acute aortic syndromes, follow-up of vascular endoprosthesis and cardiac ischemia) [42,43,44,45]. This review explores the utility of DECT in daily clinical practice in thoracic diseases, and highlights the recent state-of-the-art applications of this technique in chest pathology, focusing on its potential uses in oncology and emergency imaging (Table 1 and Table 2). Furthermore, DECT technology could represent an important part of the future of imaging innovation, especially if combined with radiomics and artificial intelligence (AI), which could further significantly increase its diagnostic potential.

## 2. Relevant Sections

### 2.1. Pleura

The correct characterization of pleural plaques or nodular-like pleural lesions represents a diagnostic challenge for radiologists, and it is a paramount aspect in establishing the most appropriate diagnostic–therapeutic process for certain findings.

Lennartz et al. showed that using iodine maps with an optimal iodine concentration IC threshold of 1.3 mg/mL improved both the quantitative and qualitative assessment of pleural carcinomatosis compared with noncalcified benign pleural lesions (Figure 1). In this study, 84 patients (40 patients with non-malignant pleural lesions and 44 patients with pathological proven pleural carcinomatosis), who underwent a post-contrast late venous-phase DETC on the chest, were retrospectively evaluated. The sensitivities for representing pleural carcinomatosis with 40 keV VMI compared to conventional CT were 96% (199 of 208) and 83% (172 of 208), respectively, with specificities of 84% (161 of 192) and 63% (120 of 192), respectively (*p* ≤ 0.001 each). Pleural carcinomatosis showed higher IC (2.3 mg/mL ± 0.8) compared to benign pleural lesions, and 1.3 mg/mL for IC was identified as the optimal threshold (sensitivity of 92% (48 of 52), specificity of 88% (42 of 48)) [46] (Figure 2 and Figure 3).

The differential diagnosis of pleural effusion may represent another considerable challenge, since in about 20–40% of patients the nature of pleural effusion remains undiagnosed, and often more invasive procedures such as thoracoscopy are needed to make a correct diagnosis. Zhang et al. retrospectively examined 29 patients, 14 with benign pleural effusion (secondary to pneumonia, tuberculosis, pneumocystosis, etc.) and 15 patients with malignant effusion (secondary to lung adenocarcinoma, squamous cell carcinoma, adenosquamous carcinoma) that underwent non-contrast DECT imaging. The authors demonstrated that the effective atomic number (threshold 7.885: AUC 0.740, sensitivity 80%, specificity 71.4%) and HU attenuation at the VMI40 keV (HU threshold 38.4: AUC 0.795, sensitivity 92.5%, specificity 53.5%) and VMI100 keV (HU threshold 6.32: AUC 0.843, sensitivity 100%, specificity 66.7%) combined with disease history and patient’s age, can provide a diagnostic tool for pleural effusion [47].

Thus, DECT could aid radiologists in the detection and characterization of pleural lesions and in defining the nature of pleural effusion, even if validation on a larger cohort is needed.

### 2.2. Lung

In the past, some published works have tried to demonstrate the diagnostic role of DECT in neoplastic lung diseases [78,79,80,81,82,83,84,85].

Sato et al. compared the diagnostic value of spectral analyses of increasing ring peripheral IC with conventional findings for differentiating primary lung tumours from pulmonary metastases. The authors retrospectively studied 93 patients who underwent a preoperative DECT and the subsequent surgical excision of primary lung carcinoma (*n* = 68) or lung metastases (*n* = 25). Lung metastases, compared with primary pulmonary tumor, showed a significantly higher frequency of ringlike peripheral high iodine concentration (52% vs. 19%) but no significant difference in frequency of spiculated margins or rim enhancement. In a multivariable analysis including lesion diameter, smoking history, and ringlike peripheral high IC, the only independent significant predictor of pulmonary metastasis was ringlike peripheral high IC (interobserver agreement, expressed as kappa, was 0.80). Thus, ringlike peripheral high iodine concentration could be a valuable aid in the evaluation of indeterminate solitary nodules [48].

He et al. evaluated DECT capabilities in the differential diagnosis between malignant and benign nodules. They reviewed 151 pathologically confirmed solid pulmonary lesions (78 lung cancers and 73 benign lesions) and evaluated diameter, volume, Lung CT Screening Reporting and Data System (Lung-RADS), and DECT spectral parameters such as effective atomic number, IC, and normalized IC in arterial and venous phases. A cut-off value of IC 0.95 mg/mL and normalized IC of 17.98% in the venous phase showed a good diagnostic performance in the identification of lung cancer (AUC: 0.891, sensitivity: 0.987, specificity: 0.753, and AUC: 0.888, sensitivity: 0.974, and specificity: 0.781, respectively). The authors concluded that the combination of morphological and quantitative features (IC and normalized IC in the venous phase) could improve diagnostic performance [49].

A multi-institutional and prospective study investigated the value of morphological criteria in non-contrast and contrast-enhanced CT, using data from perfusion CT imaging, and DECT in differentiating benign and malignant solitary pulmonary nodules. A total of 285 solitary pulmonary nodules were scanned with the different techniques and the value of normalized IC in the venous phase, with a threshold of 0.35, showed an AUC of 0.79, a sensitivity of 75.45%, and a specificity of 80.51% [50]. Ha et al. investigated the use of DECT for the differential between lung metastases and benign pulmonary nodules diagnosis in patients affected by thyroid carcinoma; the authors retrospectively studied the spectral data of 153 nodules in 63 patients (55 lung metastases and 97 benign nodules). Lesions were considered metastatic for an iodine uptake on I-131 single-photon emission CT or increased size in follow-up CT. The DECT parameters of the metastatic lesions were significantly higher than those of the benign nodules (Z-effective value, 10.0 ± 0.94 vs. 8.79 ± 0.75, IC, 5.61 ± 2.02 mg/mL vs. 1.61 ± 0.98 mg/mL; normalized IC, 0.60 ± 0.20 vs. 0.16 ± 0.11 and slope of the spectral attenuation curves (λ), 5.18 ± 2.54 vs. 2.12 ± 1.39; all *p* < 0.001). The cutoff value for IC was 3.10, for normalized ICC 0.29, and for slope HU 3.57. Thus, DECT parameters could help to differentiate metastatic and benign lung nodules in patients affected by thyroid cancer, even those measuring ≥3 mm and <5 mm, with an AUC of 0.975, 0.979, and 0.865, respectively, for IC, NIC, and λHU [51]. The λ value is calculated as the CT attenuation difference at two different energy levels (e.g., 40 and 100 keV) divided by the energy difference (e.g., 60 keV) from the spectral HU curve, according to the formula: λ = |CT40 keV − CT100 keV|/60.

Choe et al. studied if applying radiomics to DECT iodine overlay maps could predict survival outcomes in patients with resectable lung cancer. The authors evaluated 93 patients with lung cancer eligible for curative surgery. Multivariate analysis showed that stage and entropy were independent risk factors predicting overall survival and disease-free survival; they concluded that radiomics features (histogram entropy) on iodine overlay maps could add valuable prognostic information for patients with resectable lung cancer [52].

Hagen et al. [53] compared the image quality and radiation dose of contrast-enhanced chest CT in an oncologic cohort (100 patients) of a second-generation DECT and a first-generation photon-counting scan (PCD-CT). Two radiologists assessed image quality for mediastinum, vessels and lung parenchyma using a Likert scale and performed measurements of contrast-to-noise ratio (CNR), signal-to-noise ratio (SNR) on vessel and pulmonary parenchyma, and in the case of the presence of lung metastases, performed tumor-to-lung parenchyma contrast ratio. Image quality was significantly higher on PCD-CT than DECT images and, above all, the radiation dose for contrast-enhanced chest CT could be significantly reduced compared to second-generation DECT. This study describes very promising preliminary results, but photon-counting scans are still far from clinical application, and further studies are needed to evaluate reproducibility with different DECT scans technologies.

In the field of interstitial lung disease (ILDs), Chen et al. published a recent study that evaluated the role of DECT for quantitative severity assessment in connective tissue disease-associated ILDs. The authors designed a cross-sectional study to explore if DECT parameters (effective atomic number (Zeff), lung lobe volume, and monochromatic CT number of each lung lobe at the 70 keV energy level) could reflect the grade of severity of connective tissue disease ILDs evaluated by CT (CT images were reconstructed from the DECT data set) associated with some clinic data such as pulmonary function test (forced vital capacity (FVC%predicted) and diffusing capacity of the lungs for carbon monoxide (DLCO%predicted)) and symptoms scales (Borg dyspnea score, Leicester cough questionnaire, and life quality scale). They enrolled 147 adult patients with DECT scans and showed that increased severity of interstitial lung disease, assessed by high-resolution CT, symptoms, and pulmonary function test, was significantly associated with reduced lung volume (2309.51 cm^3^ vs. 3475.21 cm^3^), elevated value (3.104 vs. 2.256) of Zeff and increased monochromatic CT number (−722.87 HU vs. −802.20 HU); indeed, these parameters had a good differentiating ability to detect extensive ILD from limited disease and significant correlation with FVC% predicted. Moreover, the increased monochromatic CT number averaged (cut-off: −762.30 HU) over the whole lung had the best performance for extensive fibrotic form discrimination (AUC = 0.901), with a sensitivity of 82.1% and a specificity of 85.4%. The Zeff value was the independent risk factor for dyspnoea and cough. Thus, DECT analysis could provide a further objective tool for use in the diagnosis and monitoring of these lung diseases, although caution must be exercised in extrapolating these results, and further prospective validations of such preliminary results should be followed, without a significant imbalance in the radiological patterns (usual interstitial pneumonitis and nonspecific interstitial pneumonia) among the patients included [54].

Scharm et al. [55] recently published a retrospective study of 32 patients with the aim to evaluate if the data of ventilation, lung perfusion, and late enhancement with DECT scan can be utilized as early imaging markers for disease progression in patients with idiopathic pulmonary fibrosis (IPF), which could also be useful in assessing response to treatment with anti-fibrotic drugs. Ventilation, lung perfusion, and late enhancement images were calculated from the ventilation–perfusion–late enhancement-CT. The VNC images in inspiration were registered to the VNC images in expiration. The VNC inspiration and lung perfusion images were transformed to match the expiration scan (VNC inspiration warped and lung perfusion warped, respectively). The inverse Jacobian determinant was used to calculate the regional ventilation from the VNC images in inspiration and expiration. The inspiration scan was performed in the arterial phase and, subsequently, the expiration scan was acquired with a 300 s of delay after intravenous administration of a 60 mL contrast media (Iomeron 400, Bracco, Milan, Italy); VNC and pulmonary blood volume (PBV) images were generated. DECT scans were performed at two-time points, with a mean interval of 15.4 months. DECT parameters were compared with longitudinal changes in pulmonary function test (FVC%, DLCO%), mean lung HU density in CT, and lung volume. The authors demonstrated that regional ventilation and late enhancement at baseline preceded future changes in lung volume; moreover, regional ventilation also correlated with a future change in FVC%. Thus, considering the limit of a small cohort of patients, DECT parameters could potentially be early markers in the determination of progression in patients affected by IPF; a validation on a larger cohort would be appropriate.

### 2.3. Breast

In breast cancer, there is a wide variation in metastatic potential depending on molecular subtype, tumour size, stage, and grade. Usually, invasive breast lesions are hypervascular relative to normal surrounding parenchyma, with significant iodine uptake. Due to this aspect, the advantage of DECT would lie in the possibility of increasing the conspicuity of the primary breast lesion as part of a whole-body imaging, allowing complete staging with one single exam. Indeed, Metin et al. investigated the role of DECT in improving the conspicuity of primary breast lesions. The authors enrolled 29 patients with 39 histopathologically proven breast cancers; all examinations were performed with DECT in the late post-contrast arterial phase (50 s after injection of intravenous contrast medium). Radiologists visually graded the conspicuity of lesions on VMI and with a subjective score proved the highest conspicuity of breast cancers with VMI40 keV (Figure 4) [86,87]. In this regard, Wang et al. showed that VMI at low keV in the venous phase acquired by DECT improved the objective and subjective assessment of lesion conspicuity in patients with malignant breast lesions [56]; Li et al. demonstrated VMI 80 keV is the parameter most suitable for observing the breast and lungs simultaneously by using monochromatic spectral images, and iodine maps can be very useful in distinguishing malignant breast neoplasms from normal breast tissue, especially when breast density is greater than 25% according to Wolfe’s classification, because contrast may become sharper when breast density is higher [88].

Volterrani et al. retrospectively analyzed 31 patients (64 breast lesions) and evaluated DECT capabilities in the identification and loco-regional staging of primitive breast cancers, trying to understand if DECT was able to distinguish different histotypes. All examinations were performed with a DE scan in the late post-contrast arterial phase (45–50 s after injection of intravenous contrast medium) and VMI (40 and 70 keV), and the iodine maps were extrapolated in post-processing images. DECT identified 67 hypervascular breast lesions (all invasive cancers, 8/10 carcinomas in situ, and 5 non-malignant lesions). In about 85% of the cases, the T-stage was correctly assessed. The use of iodine maps and IC demonstrated an ability to distinguish invasive tumours from other lesions with an AUC of 0.968 (sensitivity 94.9%, specificity 93.0%, IC threshold > 1.70 mg/mL). The ratio of the CI of breast lesions to the CI of normal breast parenchyma showed that a threshold > 6.13 was the best for distinguishing invasive ductal carcinoma from other lesions (AUC value, 0.914; specificity, 81.1%; sensitivity, 87.0%) [57].

Moon et al. tried to correlate tumor conspicuity on VMI with some prognostic histopathological features in 64 patients with breast cancers. All examinations were performed with a DE scan in the post-contrast arterial phase (controlled using bolus tracking) and delayed (90 s after injection of intravenous contrast medium injected via a peripheral vein at a rate of 2/2.5 mL/s, Omnipaque 350). The authors found that VMI 40 keV of delayed phase yielded the highest conspicuity and attenuation values for cancers with the greatest AUC (AUC 0.817); however, attenuation values and conspicuity scores were significantly higher on arterial phase in HER-2 Enriched and Triple Negative subtypes (high histologic grade, ER -, PgR -, HER-2 + and Ki67 high). In the delayed phase, no significant difference was found between low- and high-risk tumors, probably due to the arterial phase best reflecting the difference in neoangiogenesis, and because in the delayed phase the washout should have already occurred [58].

A recent study would suggest that radiomic features of the iodine map could provide potentially useful information on tumor biology regarding intrinsic differences between primary and metastatic tumor tissue that could be useful for therapeutic purposes, as well as the noninvasive prediction of the presence of distant metastases. Lenga et al. enrolled and retrospectively studied 77 treatment-naïve patients with biopsy-proven breast carcinomas (41 nonmetastatic, 36 metastatic). Following principal component analysis, a multilayer perceptron artificial neural network (MLP-NN) was used for classification (70% of cases for training, 30% validation). Histopathology served as a reference standard. MLP-NN predicted metastatic status with AUCs of up to 0.94, and accuracies of up to 92.6 in the training and 82.6 in the validation datasets (accuracy training: mean 75.78%, median 75.9% interquartile range 74.1–76.83%; accuracy validation: mean 73.92%, median 73.9%, interquartile range 69.6–78.3%). The separation of the primary tumor and metastatic tissue yielded AUCs of up to 0.87, with accuracies of up to 82.8 in the training and 85.7 in the validation dataset (accuracy training: mean 74.78%, median 74.75%, interquartile range 72.8–77.08%; accuracy validation: mean 72.87%, median 73.2%, interquartile range 69–77.8%) [59].

### 2.4. Lymph Nodes

Correct imaging characterization of sentinel lymph nodes in preoperative breast cancer staging is paramount for optimal therapeutic planning. Zhang et al. [60] from June 2015 to December 2017 evaluated 193 patients; all examinations were performed with DE scans in dual post-contrast phases. Two radiologists analyzed lymph nodes’ morphological and quantitative spectral parameters, such as the IC and the λHu of the spectral curve (normalized to those of the aorta in each phase) in the difference between 70 and 40 keV. Histopathologic analysis revealed that 55 sentinel lymph nodes were metastatic and 138 were uninvolved.

Among the quantitative spectral parameters, the λHu in the venous phase showed the highest diagnostic accuracy (90.5%, sensitivity 66%, specificity 97.7%, AUC of 0.88) (Figure 5 and Figure 6). Among the morphological criteria, the cortex status of the axillary lymph node (abnormal vs. normal) obtained the highest accuracy (81%), with high specificity (93.6%), but unfortunately with an AUC of 0.62 and very low sensitivity (31%).

Recently, Li et al. showed that the spectral quantitative parameters derived from contrast-enhanced dual-layer CT, such as λHu in the delayed phase, can be applied for the assessment and preoperative detection of axillary lymph node metastasis in breast cancer with an AUC of 0.93, a sensitivity of 92.3%, and a specificity of 87.5% [89].

Terada et al. retrospectively analyzed 137 patients (39 lymph nodes metastases and 98 lymph nodes non-metastases) and evaluated relationships between lymph node (LN) metastasis and simple DECT parameters, with pathological (nuclear grade, estrogen receptor status, and Ki67 index) and morphological features (shortest and longest diameters of the lymph nodes, longest-to-shortest diameter ratio > 2:1, and hilum). They concluded that the delayed-phase DECT parameters (scan at 120 s after the start of the contrast injection), such as the λHU, IC, and attenuation values at 40 keV and 70 keV, could be an important tool for predicting lymph nodes metastasis, regardless of LN size. However, DECT parameters could be influenced by inner differences in the DECT scanner, but also in the scanning protocols and injection protocols of the iodinated contrast medium [61].

Therefore, dual-energy CT appears to be a very promising tool for the preoperative evaluation of sentinel lymph nodes in breast cancer patients.

Some morphological criteria for conventional CT analyses are commonly used to assess if neoplastic lymph node involvement is present, such as size (diameter of the short axis ≥ 10 mm) and necrotic appearance [90].

Yang et al. demonstrated that quantitative parameters exhibit greater accuracy than conventional CT morphological assessment. Authors retrospectively studied 84 patients with suspected lung cancer; a total of 144 lymph nodes were included, with 48 metastatic lymph nodes and 96 non-metastatic lymph nodes. They underwent chest DECT dual-phase post-contrast scans. Spectral parameters were analyzed. The λHU curve measured during both arterial and venous phases was significantly higher in metastatic lymph nodes than in benign lymph nodes. When the optimal threshold value of λHU was 2.75, the overall accuracy in the diagnosis of metastatic lymph nodes was 87.0% [37].

Sekiguchi et al. demonstrated that VMI40 keV performed about 60 s after the iodinated contrast medium injection could be useful for the correct evaluation of the hilar lymph nodes at the pulmonary level, given the greater intrinsic contrast differences between the pulmonary vessels and hilar lymph nodes [91].

Nagano et al. [62] performed a retrospective comparative study that aimed to evaluate the diagnostic accuracy of DECT and FDG PET/CT for metastatic mediastinal lymph nodes in patients with non-small-cell lung cancer in a cohort of 57 patients who underwent a preoperative DECT and PET/CT and subsequent and mediastinal lymph nodes resection. Two radiologists independently reviewed images, and evaluated the morphological criteria (long and short axis, presence of necrosis) and electron density (ED) images. The sensitivity, specificity, and accuracy for nodal metastasis were 54.5%, 85.7%, and 76.9% for a short axis diameter greater than 8.5 mm, respectively; 63.6%, 73.8%, and 70.9% for a long axis diameter greater than 13.0 mm; 15.2%, 98.8%, and 75.2% for the presence of necrosis; 51.5%, 79.8%, and 71.8% for attenuation on 120 kVp images of 95.8 HU or less; 87.9%, 58.3%, and 66.7% for ED of 3.48 × 1023/cm3 or less; and 66.7%, 75.0%, and 72.6% for positive FDG uptake. Accuracy was highest for the combination of ED measurement and short axis diameter (accuracy, 82.9%; sensitivity, 54.5%; specificity, 94.0%) and the combination of ED and positive FDG uptake (accuracy, 82.1%; sensitivity, 60.6%; specificity, 90.5%); these accuracies were greater than those for the individual features (*p* < 0.05). Thus, ED analysis could improve diagnostic accuracy, and may complement morphological CT criteria and FDG uptake on PET/CT in diagnosing metastatic lymph nodes. It would be appropriate and useful to also evaluate these results with other types of measurements performed on dual-source DECT maps, or rapid kVp switching or twin-beam DECT; moreover, it would be interesting to evaluate how much radiomics can really improve diagnostic accuracy with such available data.

### 2.5. Mediastinal Neoplasms

Anterior mediastinal lesions are relatively uncommon, and their differential diagnoses have been an everlasting problem for radiologists; thymoma is the most common primary neoplasm, yet it includes a large variety of entities of different diseases such as lymphoma, teratoma, simple thymic cyst, true thymic hyperplasia, thymolipoma, germ cell tumours, and seminoma. Although some specific features could aid radiologists in try defining hypotheses of a pathological nature, such as the presence of intralesional fat, hyperdense tissue at unenhanced CT, vivid post-contrast enhancement, and cystic or necrotic component, diagnosis is often hard; in this context, DECT can facilitate the diagnostic process. Both lymph node and pleural metastases may be present in thymic carcinoma, and therefore the crura of the diaphragm should be carefully analyzed. Diagnosis and correct staging are crucial for treatment modalities; indeed, low-risk or early-stage thymomas (World Health Organization classification A, AB, and B1 or Masaoka–Koga stage I and II) are usually treated with surgery alone, whereas advanced stages currently require a multimodal approach with surgery and/or chemoradiation [63,92,93,94,95].

Yan et al. [96] evaluated 57 patients affected by thymic carcinoma (*n* = 14), thymic lymphoma (*n* = 12), low-risk thymoma (*n* = 16), and high-risk thymoma (*n* = 15). All examinations were performed using dual-phase DECT, arterial and venous, spectral post-processing with extrapolation and specific analysis of VMI and iodine maps. A cut-off value of 1.25 mg/mL at IC in the venous phase yielded the best possible accuracy in differentiating low-risk thymomas from thymic lymphoma, with an AUC of 0.969, a sensitivity of 87.5%, and a specificity of 100%. In the venous phase, a cut-off of iodine-related HU of 34.3 HU (a quantitative parameter on colour-coded iodine overlay maps that represents a new metric of tissue iodine up-take in Hounsfield units as a surrogate measure of the conventional analysis with conventional CT) showed the best accuracy in differentiating low-risk thymomas from high-risk thymomas/thymic carcinoma with an AUC of 0.893, a sensitivity of 75.0%, and a specificity of 89.7% (Figure 7 and Figure 8).

Yu C. et al. evaluated 88 patients with different WHO types of thymoma who underwent DECT perfusion scans (*n* = 51) and conventional CT enhancement scans (*n* = 37). The authors found that perfusion and spectral parameters of thymoma types A and AB were significantly higher than those of other subtypes, due to their different histopathological composition [97].

Zhou et al. analyzed 56 patients and explored the utility of DECT in distinguishing thymic epithelial tumours (*n* = 35) from thymic cysts (*n* = 21), with a diameter < 5 cm, pathologically confirmed (Figure 9). In the arterial phase, analyzing via VMI60 keV with a cut-off value of 68.42 HU demonstrated a good diagnostic performance in differential diagnosis with an AUC of 0.978; otherwise, the same analysis in the venous phase VMI70 keV with a cut-off value of 59.77 HU achieved a good diagnostic performance in differential diagnosis, with an AUC of 0.956. The IC evaluation with cut-off values of 10.99 mg/dL and 11.05 mg/dL, used to distinguish small neoplasms from thymic cysts, presented an AUC of 0.956 and 0.924, respectively [64]. Rajamohan N. et al. revealed several parameters with excellent diagnostic performances in CT texture analysis for differentiating thymoma from thymic hyperplasia, low from high thymoma (WHO grade), and low from high Masaoka Koga/International Thymic Malignancy Interest Group stages [98].

### 2.6. Esophagus

Recently, some studies have been published that try to evaluate a possible diagnostic role in esophageal cancer. Even if some results would seem encouraging, data in the literature and experience in this field are still limited. Consequently, further evidence needs to be validated. Cheng et al. retrospectively evaluated 68 patients and concluded that low-keV monoenergetic reconstructions could improve the assessment of lesion conspicuity, and may also have great potential for preoperative T-evaluation in patients with esophageal cancer [65]. Zopfs et al. [66] investigated the diagnostic value of low-keV monoenergetic reconstructions and iodine overlays for the locoregional pre-therapeutic assessment of esophageal neoplasm. They evaluated 74 patients and showed that VMI 40–60 keV (peaking at VMI40 keV) had significantly higher attenuation and signal-to-noise ratio than conventional CT for all ROIs evaluated, while iodine overlays and VMI40-60 keV could improve the qualitative assessment and conspicuity of the esophageal tumor (Figure 10).

### 2.7. Vascular Emergency

In recent years, several studies have been published in the literature that have demonstrated and confirmed the diagnostic quality of DECT in the emergency scenario, and the use of VMI reconstructions at 70 keV is preferred as the optimal energy level for chest CT angiography images, while additional reconstructions at 40 or 50 keV may be useful when poor vascular contrast is noted [99,100,101,102,103].

A recent study tried to assess if true non-contrast (TNC) could be replaced by VNC for aortic intramural hematoma (IMH) diagnosis in acute aortic syndrome in a phantom and clinical evaluation. Patients with confirmed IMH and who underwent a dual-phase with a dual-layer DECT were included retrospectively in two different centers. For in vitro imaging, a custom-made phantom of IMH was placed in a semi-anthropomorphic thorax phantom; for in vivo imaging, 21 patients were enrolled. The authors demonstrated that there is no significant difference in diagnostic image quality between VNC and TNC images on dual-layer DECT, and VNC offers similar diagnostic performances for IMH evaluation, with a potential decrease in the radiation dose delivered to the patients (Figure 11) [67].

The early diagnosis of pulmonary embolism is crucial for better clinical outcomes, and one of the most commonly accepted indications of DECT is the study of pulmonary circulation. Effectively, DECT could lead to greater confidence in the interpretation of pulmonary angiographic CT images due to the simultaneous assessment of pulmonary vasculature and parenchymal iodine distribution through the use of iodine maps, which allows for the assessment of any pulmonary perfusion defects (Figure 12) [42,68].

A recent meta-analysis [69] evaluated the accuracy of DECT in the diagnosis of pulmonary embolism. Seven studies entered the analysis and, of the 182 patients included, 108 patients had pulmonary embolisms. The pooled analysis showed an overall sensitivity and specificity of 88.9% and 94.6%, respectively; Cochran-Q was 0.8712, and AUC was 0.935 in the ROC curve. The authors showed high sensitivity, specificity, and diagnostic accuracy in the diagnosis of acute pulmonary embolism with a high positive likelihood ratio; however, they concluded that studies with a larger cohort of patients and with standardized reference tests are still needed. Another meta-analysis [70], conducted according to PRISMA, investigated the diagnostic performance of DECT, with regard to its post-processing techniques, namely, linear blending, iodine maps, and VMI, in diagnosing acute pulmonary embolism. The authors included seventeen studies in the analysis, and subsegmental acute pulmonary embolism was excluded from the analysis due to data heterogeneity or lack of data. The pooled analysis showed sensitivity and specificity of 0.87 and 0.93 for linear blending alone, 0.89 and 0.90 for linear blending and iodine maps, and 0.90 and 0.90 for linear blending, iodine maps, and virtual monoenergetic reconstructions. The authors concluded that, despite the high values reported, the diagnostic performance of DECT is not superior to single-energy CT (sensitivity 0.83 and specificity 0.96).

Fang Huang et al. investigated preoperative diagnostic performance in identifying the clinical rupture site of a thoracic aortic dissection using dual-source computed tomography. The authors retrospectively analyzed 150 patients with suspected thoracic aortic dissection from January 2014 to October 2017. The sensitivity of dual-source CT, conventional CT, and transthoracic echocardiography for locating the rupture site of the dissection was 100%, 93.5%, and 89.5%, respectively, and the specificity was 100%, 88.9%, and 81.5%. They concluded that the accuracy of DECT for assessing rupture sites in thoracic aortic dissection was similar to that of the gold standard, and its diagnostic sensitivity and specificity were higher than those of conventional CT and transthoracic echocardiography [104].

Flors et al. [71] retrospectively evaluated the diagnostic value of DECT in the detection of endoleaks in patients with thoracic aortic aneurysm who have undergone thoracic endovascular aortic repair (TEVAR), and investigated if a double-phase or a single delayed-phase DECT scan (300 s after the beginning of injection of intravenous contrast medium, optimal for the detection of low flow endoleaks) could replace the standard triphasic protocol (in inspiratory breath-hold: unenhanced scan, an arterial phase, and a delayed phase). Three different reading sessions were performed.

In session A, true non-contrast, arterial phase, and dual-energy delayed-phase weighted-average image datasets were read (reference standards for the diagnosis of endoleak). In session B, VNC and the dual-energy delayed phase datasets (80-kVp, 140-kVp, and weighted average images) were administered to the readers. In session C, immediately after reviewing and recording the final diagnosis for each case in reading session B, arterial phase images were added. Session B had 100% specificity, 85.7% sensitivity, 94.6% positive predictive value (PPV), and 100% negative predictive value (NPV). The two-phase study (session C) was characterized by 100% of specificity, sensitivity, VPN, and PPV. The authors concluded that VNC and delayed single contrast-enhanced DECT acquisition could replace the standard triple-phase protocol in follow-up examinations after TEVAR, with a significant dose reduction.

Many studies have investigated the benefits of DECT in the evaluation of patients affected by a thoracic aortic aneurysm in terms of dose reduction, but above all in terms of diagnostic quality and a reduction in the volume of contrast medium injected, up to 70%, guaranteeing similar aortic attenuation, signal-to-noise ratio, and contrast-to-noise ratio when compared to standard single-energy CT aortography in the same patient [72,105].

A recent study demonstrated that DECT-derived iodine concentration showed better diagnostic performance than the conventional HU evaluation in early-phase cardiac CT in detecting left atrial appendage thrombus and differentiating the thrombus from circulatory stasis [106].

Ascenti et al. [107] investigated the diagnostic performance of DECT with iodine quantification in distinguishing bland from neoplastic portal vein thrombosis in patients with hepatocellular carcinoma, compared to conventional enhancement measurements. Enhancement measurement resulted in a specificity of 85.7%, a sensitivity of 92.3%, a negative predictive value of 94.7%, and a positive predictive value of 80%. An iodine concentration of 0.9 mg/mL optimized discrimination between neoplastic and bland thrombosis with an AUC of 0.993, a sensitivity of 100%, specificity of 95.2%, a negative predictive value of 100%, and a positive predictive value of 92.9% (Figure 13). Thus, iodine quantification could be a useful tool to improve the characterization of portal vein thrombosis or other veins involved. Thrombosis of the vena cava and the atrium is frequently seen in renal cell carcinoma.

### 2.8. Bone

The potential aid of DECT in the diagnosis and follow-up of multiple myeloma has been demonstrated and recognized in the literature, and it is a reasonable and cost-effective initial imaging approach compared with whole-body MRI [73,74]. A promising study that could further clarify the role of DECT compared to MRI may be the ongoing study “Dual-Energy Computed Tomography for Improving Imaging Assessment of Multiple Myeloma (DECIMA)”. This trial aims to evaluate the sensitivity and specificity of DECT in a cohort of untreated patients, and compare the performance of DECT with standard simulated CT (data extrapolated by DECT) and whole-body MRI with a reference standard along with bone marrow biopsy results. Compared to MRI, DECT would guarantee shorter scan times (sometimes long MRI examinations would be counterproductive for patients with mixed pain from bone lytic lesions), lower costs, and shorter waiting lists. Bone is the third most common site for metastasis in cancer patients, and the detection of skeletal metastasis is therefore crucial for radiologists and clinicians to avoid painful sequelae that can lead to an inability for patients. The role of DECT in the diagnosis of bone metastases is not well defined, and there is no consensus since there are few retrospective studies in the literature, performed on a small cohorts of patients. Skeletal metastases can be pure osteolytic or osteoblastic lesions, but often they are mixed or within the bone marrow (Figure 14) [75]. Yue et al. used postprocessing, and specifically VMI, to improve the detection of subtle osteoblastic metastases of the vertebra from lung neoplasm. The authors demonstrated that VMI 70 keV could be the best tool for diagnosing not clearly evident vertebral metastases, which may have a density very similar to neighboring bone tissue [108]. A recent review [109] showed that VNC images have been used for detecting bone marrow oedema in vertebral fractures, and could increase lesion attenuation. Moreover, iodine maps could increase iodine density in metastatic bone lesions compared to that in normal bone (with a sensitivity of 90.7% with a threshold of iodine density of 4.5 mg.mL^−1^), and hydroxyapatite–water material decomposition could improve the diagnosis of bone marrow metastases, especially for subtle isodense tumors; however, age and bone mineral density could have an impact on iodine density measurements (Figure 15) [76,77,110]. Radiomic analysis could improve diagnostic accuracy and facilitate the detection of occult lesions.

## 3. Conclusions

Even if DECT is not free from pitfalls and limitations, it is progressively entering the clinical practice of every radiologist. Unlike photon-counting CT, DECT scanners are now much more widespread than a few years ago, and many radiologists are now starting to approach this new technology. Only an accurate knowledge of the basis and limits of DECT, supported by “ad hoc” courses and webinars or congresses, could guarantee optimal clinical use. The well-known benefits allow us to obtain a more confident diagnosis in emergencies and elective scenarios. Furthermore, the advantages derived from the potential reduction of the total radiant exposure and the volume of contrast media injected must be kept in mind, and should not be underestimated, especially in cancer patients or elderly people, who often use multiple potentially nephrotoxic drugs. Obviously, the application of radiomics on DECT images could make a further contribution to the diagnostic improvement of this technique. Recently, many interesting studies have been published in the literature on this topic, although more research is needed.

## Figures and Tables

**Figure 1 diagnostics-13-02440-f001:**
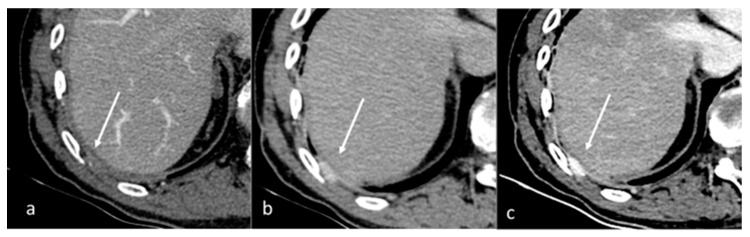
Qualitative evaluation of pleural carcinosis using an arterial phase with a conventional CT scan (**a**) and DECT scan with virtual monoenergetic reconstruction at 80 keV (**b**) and 40 keV (**c**). Low-energy monoenergetic reconstructions provide better conspicuity of the pleural lesion. The white arrow indicates the pleural metastasis.

**Figure 2 diagnostics-13-02440-f002:**
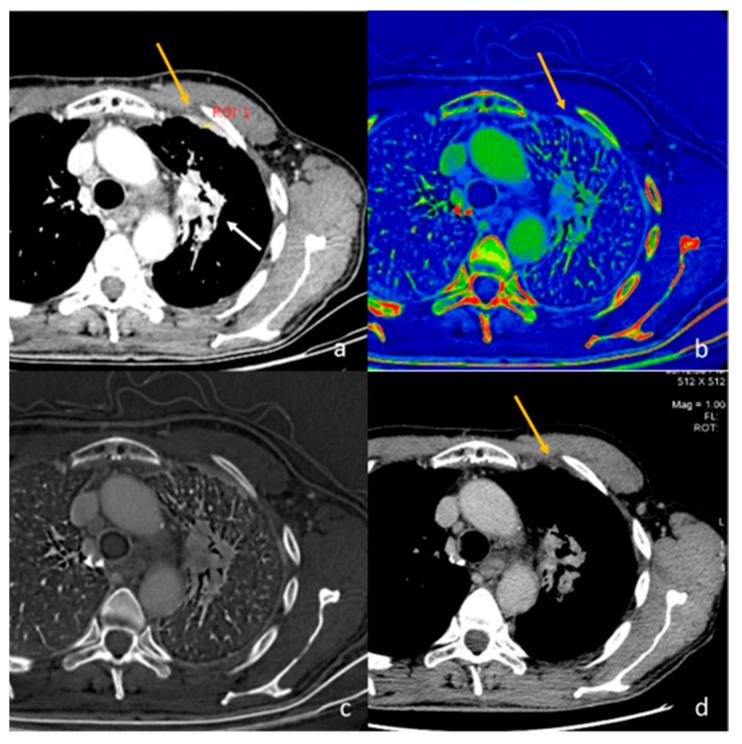
Evaluation with 40 keV monoenergetic reconstruction (**a**) and iodine map (**b**) of a pleural lesion (yellow arrow) found in the CT scan of a patient with hilar lung neoplasia (a white arrow). The pleural lesion has an iodine concentration of 1.2 mg/mL (**c**), which is therefore compatible with a benign lesion [46]. The benignity of the lesion is demonstrated by the spontaneous regression of the lesion after 2 months (**d**).

**Figure 3 diagnostics-13-02440-f003:**
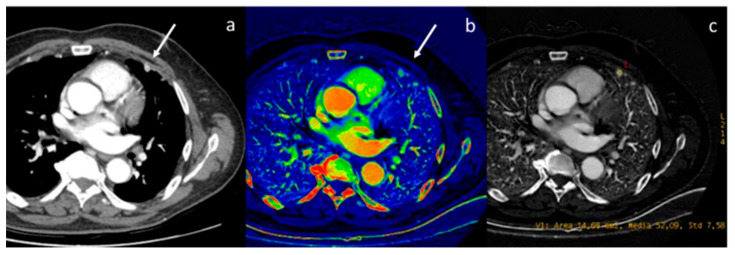
Evaluation of pleural carcinomatosis (white arrow) secondary to clear cell renal cancer with monoenergetic reconstruction at 40 keV (**a**) and iodine map (**b**). The iodine concentration is 5.2 mg/mL (**c**), thus indicative of the malignancy of the finding [46].

**Figure 4 diagnostics-13-02440-f004:**
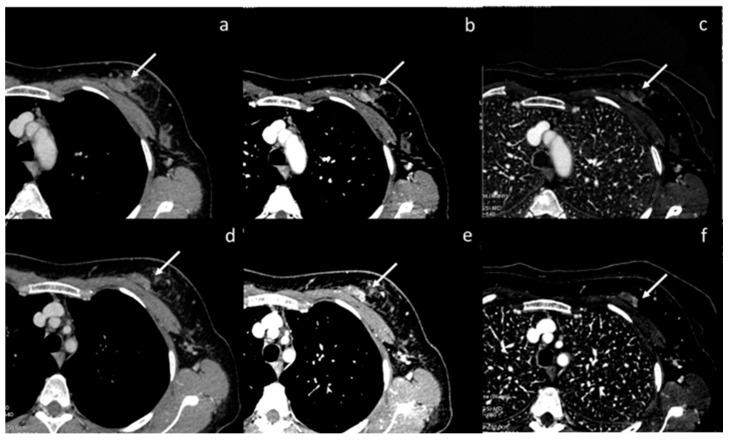
Assessment of breast neoplasia (white arrows) with monoenergetic reconstructions at 80 keV (**a**,**d**), 40 keV (**b**,**e**), and iodine map (**c**,**f**). Low monoenergetic reconstructions and iodine maps provide the highest conspicuity.

**Figure 5 diagnostics-13-02440-f005:**
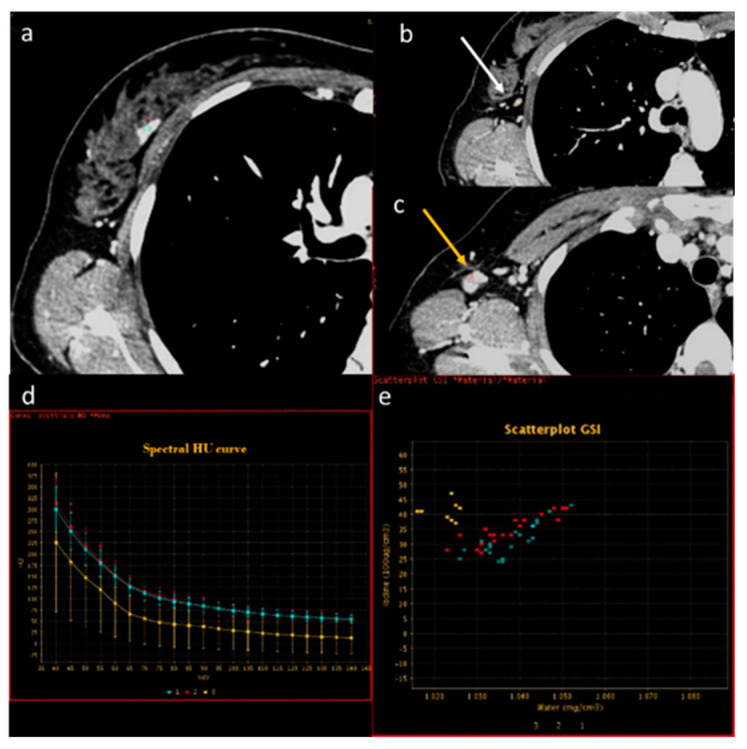
In a case of breast cancer (**a**), DECT analysis allows us to distinguish between a normal lymph node ((**b**), white arrow) and a pathological lymph node ((**c**), yellow arrow). In fact, the slope HU values (**d**) of a pathological lymph node (red ROI) and breast cancer (blue ROI) were similar if compared to normal lymph node (yellow ROI). In iodine concentration evaluation with scatterplot (**e**), there is a complete separation between breast cancer (blue ROI) and pathological lymph node (red ROI) compared to normal lymph node (yellow ROI).

**Figure 6 diagnostics-13-02440-f006:**
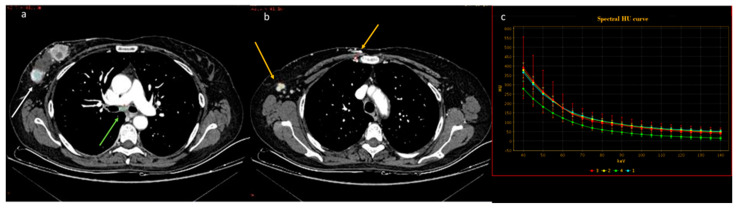
In a case of breast cancer (**a**), DECT analysis allows for distinguishing between a normal mediastinal lymph node ((**a**), green arrow and green ROI) and metastatic axillary and internal mammary lymph nodes ((**b**), yellow arrow and yellow ROI). Pathological lymph nodes present a slope of spectral HU curve (**c**) perfectly matching that of breast cancer ((**a**), white arrow, blue ROI).

**Figure 7 diagnostics-13-02440-f007:**
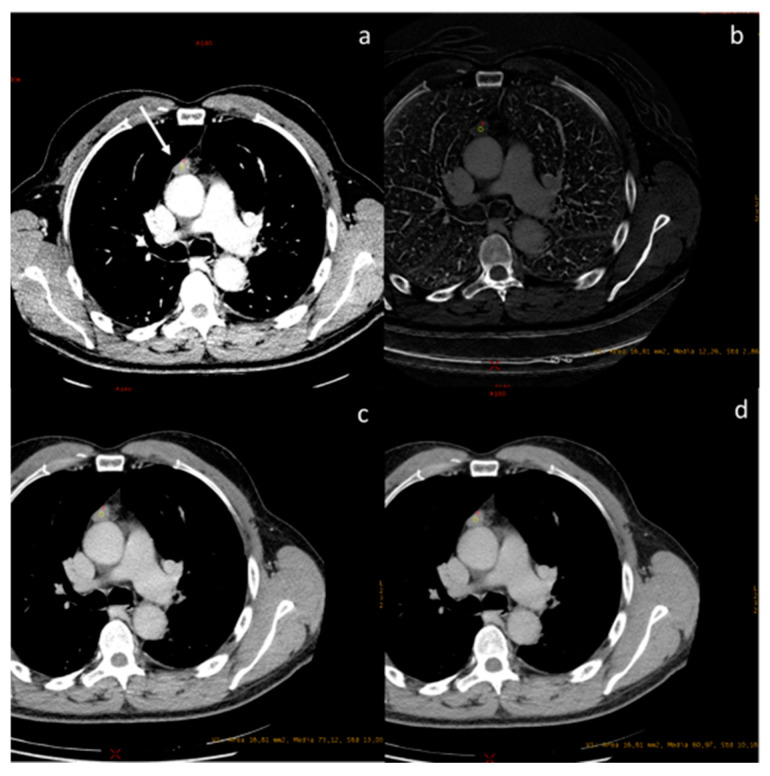
A case of spindle cell thymoma, type A, stage IIB according to Masaoka ((**a**), white arrow) with an iodine concentration of 1.22 mg/dL (**b**), and a density at 60 keV of 73.12 HU (**c**) and at 70 keV of 60.97 HU (**d**).

**Figure 8 diagnostics-13-02440-f008:**
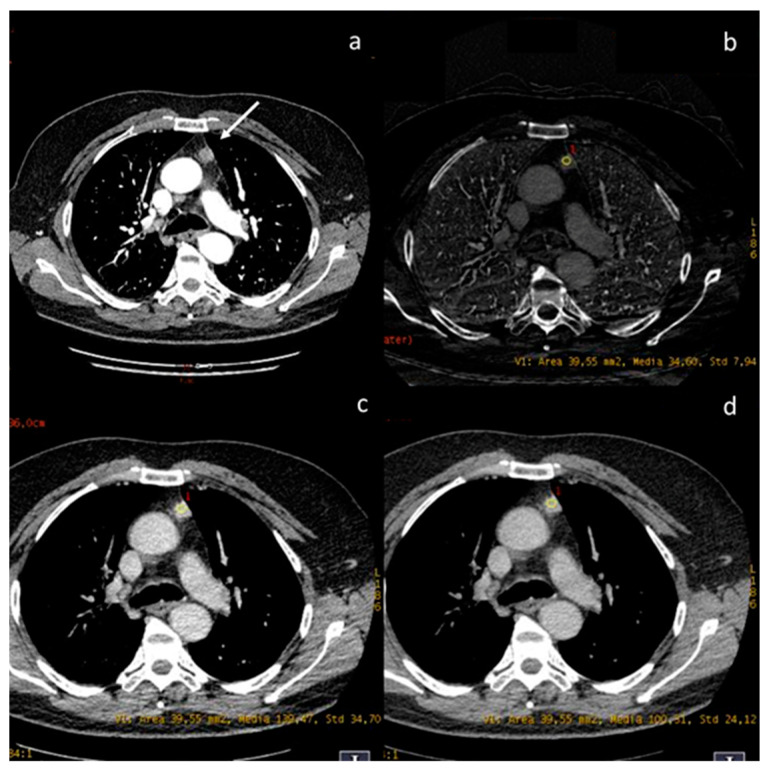
A case of thymic basaloid carcinoma ((**a**), white arrow). The iodine concentration is 3.4 mg/mL (**b**), the density at 60 keV is 139.47 HU (**c**), and at 70 keV is 100.31 HU (**d**).

**Figure 9 diagnostics-13-02440-f009:**
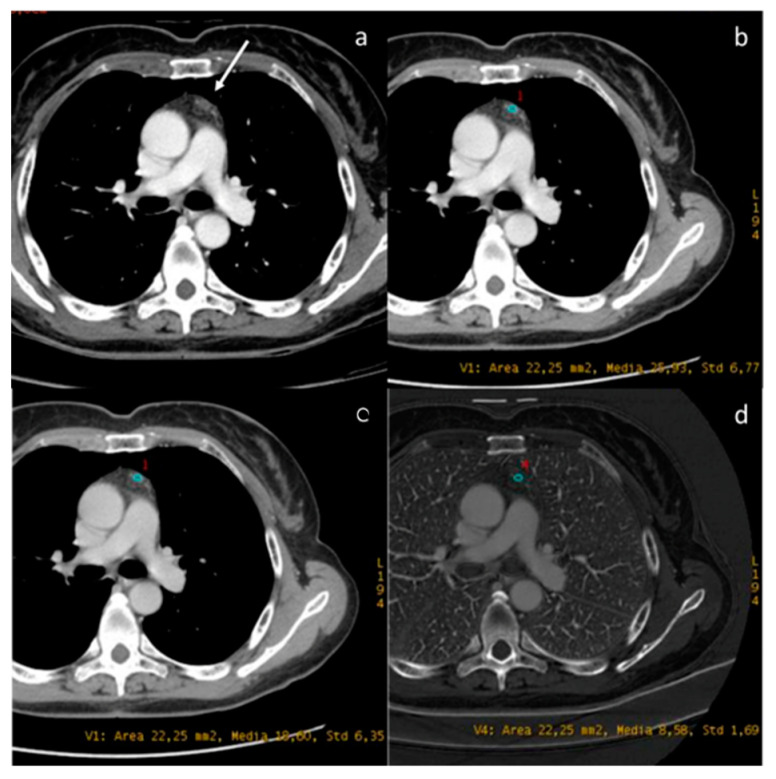
DECT provides the characterization of a thymic cyst ((**a**) white arrow). The density at 60 keV is 25.93 HU (**b**), while that at 70 keV is 18.60 HU (**c**), and the iodine concentration is 0.85 mg/dL (**d**). All values are below the cut-offs (according to Zhou et al.) to distinguish it from hypovascular solid tissue [97].

**Figure 10 diagnostics-13-02440-f010:**
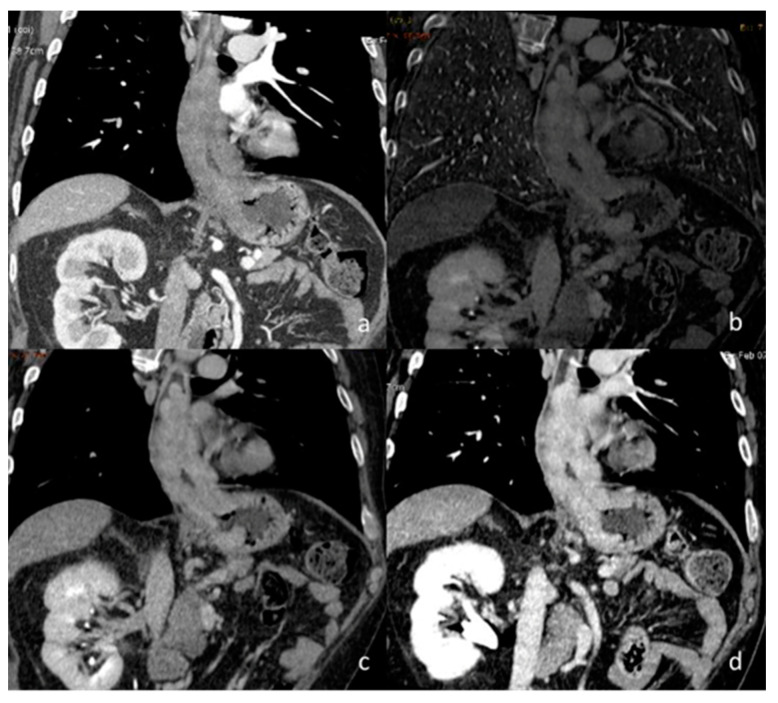
A case of esophageal adenocarcinoma with extension beyond the gastro-esophageal junction, evaluated with conventional CT in the arterial phase (**a**) and DECT in the delayed phase with iodine map (**b**) and monoenergetic reconstructions at 60 keV (**c**) and 40 keV (**d**).

**Figure 11 diagnostics-13-02440-f011:**
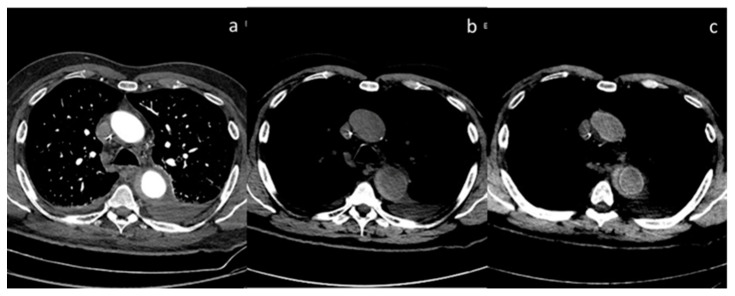
Case of aortic intramural hematoma (**a**), in which true non-contrast (**b**) and virtual non-contrast (**c**) both demonstrate efficacy in detecting intrinsic hyperintensity of the hematoma.

**Figure 12 diagnostics-13-02440-f012:**
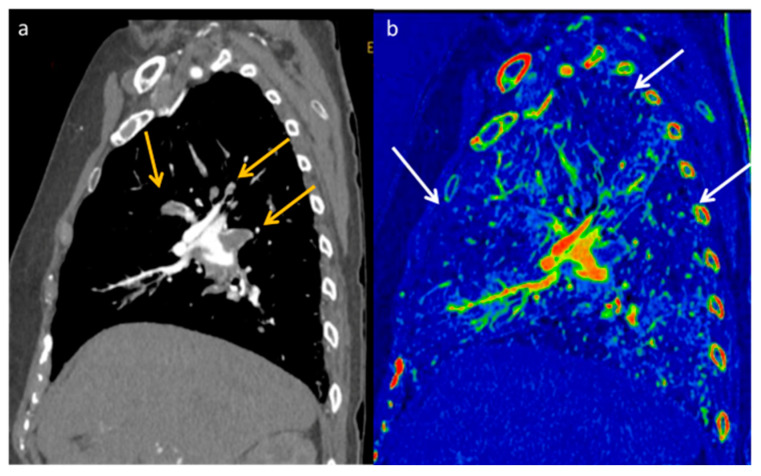
DECT allows increased confidence in detecting the presence of pulmonary embolism (yellow arrows) thanks to the increased contrast resolution provided by low-energy VMI (**a**), but it also provides the opportunity to assess the parenchymal distribution of iodine by identifying the corresponding lung perfusion defects on the iodine map ((**b**) white arrows).

**Figure 13 diagnostics-13-02440-f013:**
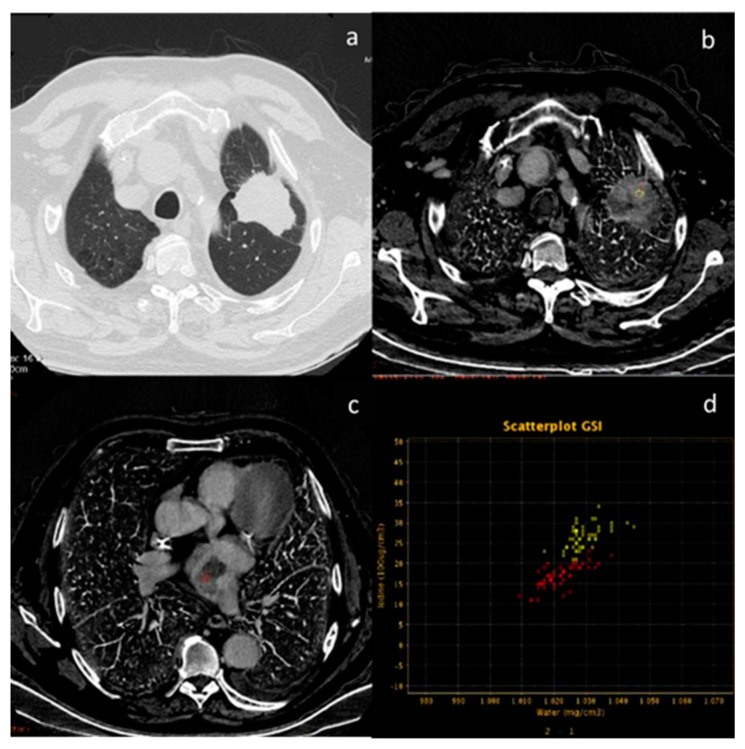
In this patient with lung cancer (**a**), neoplastic thrombosis is documented in the right atrium. The diagnosis is confirmed by the evidence of iodine concentration in both the primary neoplasm (**b**) and the thrombotic formation (**c**). The analysis is performed by scatterplot (**d**) yellow ROI neoplasm and red ROI neoplastic thrombus present a similar pathologic iodine concentration).

**Figure 14 diagnostics-13-02440-f014:**
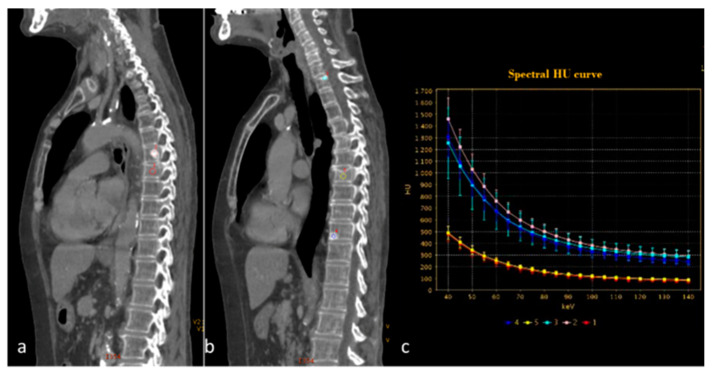
Spectral curve analysis allows a distinction to be made between osteosclerotic metastases ((**a**,**b**) pink, blue and light blue ROIs) and porotic vertebral fracture ((**a**,**b**) red and yellow ROIs). The spectral curve (**c**) demonstrates a perfect separation of the attenuation profiles between the two findings.

**Figure 15 diagnostics-13-02440-f015:**
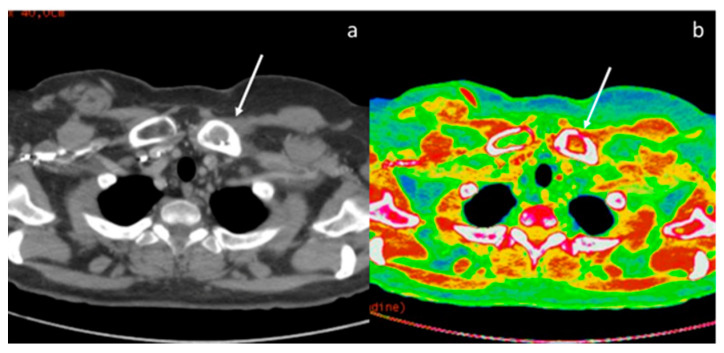
The iodine map (**b**) increases the visibility of a metastasis of the left clavicle compared to conventional CT scan with polychromatic beam (**a**).

**Table 1 diagnostics-13-02440-t001:** Description of the methodology of research in the literature.

Items	
Date of search	15/01/2023
Sources	Pubmed, Cochrane
Search terms	(dual energy CT OR dual-energy CT OR dual-energy computed tomography OR dual-energy computed tomography OR DECT) and (pleura OR lung OR breast OR mediastinum OR thymus OR embolism OR bone metastasis OR esophagus OR esophageal cancer)
Timeframe	No restrictions
Inclusion criteria	Articles published in English
Selection process	Two authors selected articles according to the following criteria: priority has been given to meta-analysis, systematic reviews, and original articles. Only original articles with diagnostic accuracy or specific dual energy measures (e.g., iodine concentration) reported in the results were included. Narrative reviews were reported only if a few original articles and no systematic reviews or meta-analysis were available about oncologic thoracic diseases.

**Table 2 diagnostics-13-02440-t002:** Main characteristics of representative studies investigating DECT in thoracic disease.

Authors and Study Design	Country	Aim/Rationale	No. of Patients	DECT Scan and Conflict of Interests	Main Conclusion
Lennartz et al. [46] Retrospective study	Germany, USA	To evaluate the use of spectral CT for differentiation between noncalcified benign pleural lesions and pleural carcinomatosis	84	IQon; Philips Healthcare, Best, the Netherlands No C.I.	Iodine overlay images and quantitative iodine maps improve the differentiation of noncalcified benign pleural lesions from pleural carcinomatosis compared with conventional CT. The benefit of spectral CT for diagnosis of pleural lesions was greater for less-experienced radiologists compared with experienced radiologists.
Zhang et al. [47] Retrospective study	China	To investigate the value of spectral CT in the differential diagnosis of benign from malignant pleural effusion	29	Discovery CT750 HD; GE Healthcare No C.I.	The CT value measurement at both high and low energy levels and the effective atomic number obtained in a single spectral CT scan can assist the differential diagnosis of benign from malignant pleural effusion. Combining them with patient age and disease history can further improve diagnostic accuracy.
Sato et al. [48] Retrospective study	Japan	To compare the utility of ringlike peripheral increased IC and conventional findings for differentiating primary lung cancers from pulmonary metastases on DECT	93	Discovery CT750 HD or Revolution HD, GE Healthcare. No C.I.	Ringlike peripheral high IC had excellent interobserver agreement and high specificity (but poor sensitivity) for differentiating pulmonary metastasis from primary lung cancer and was independently predictive of pulmonary metastasis.
He et al. [49] Retrospective study	China	To investigate the ability of quantitative parameters of dual-energy computed tomography (DECT) and nodule size for differentiation between lung cancers and benign lesions in solid pulmonary nodules	147	Somatom Definition Flash, Siemens Healthcare, Germany. No C.I.	The DECT-derived IC V and NIC V may be useful in differentiating lung cancers from benign lesions in solid pulmonary nodules.
Yan et al. [50] Multi-institutional and prospective study	China	To investigate the value of non-contrast-enhanced CT, contrast-enhanced CT, CT perfusion imaging, and dual- energy CT used for differentiating benign and malignant SPNs with a multi-institutional and prospective study	285	Revolution CT, GE Healthcare, Milwaukee WI, USA). No C.I.	SPNs evaluated with multimodality CT imaging contribute to improving the diagnostic accuracy of benign and malignant SPNs. DECT using the parameter of NIC at the venous phase is helpful for improving the diagnostic performance.
Ha et al. [51] Retrospective study	South Korea	To explore the importance of quantitative characteristics of DECT between pulmonary metastasis and benign lung nodules in thyroid cancer	63	Philips IQon 128-slice dual-layer detector spectral CT scanner. No C.I.	DECT parameters can help to differentiate metastatic and benign lung nodules in thyroid cancer. The highest diagnostic accuracy was achieved with the NIC and IC, followed by the NIC _PA_ and λHU, and their cutoff values were 0.29, 3.10, 0.28, and 3.57, respectively.
Choe et al. [52] Retrospective study	Korea	To investigate whether radiomics on iodine overlay maps from DECT can predict survival outcomes in patients with resectable lung cancer	93	Somatom Definition, Siemens Healthcare, Germany. No C.I.	Radiomic features extracted from iodine overlay map reflecting heterogeneity of tumor perfusion could add prognostic information for patients with resectable lung cancer.
Hagen et al. [53] Prospective study	Germany	To compare the image quality and the patient dose of contrast-enhanced oncologic chest-CT of a first-generation photon-counting CT and a second-generation dual-source dual-energy CT using comparable exam protocol settings	100	Somatom Definition Flash, Siemens Healthcare, Germany. NAEOTOM Alpha, Siemens Healthineers, Forchheim, Germany. Some Authors received institutional research support from Siemens	Photon-counting CT enables oncologic chest-CT with a significantly reduced dose while retaining image quality similar to a second-generation dual-source DECT.
Chen et al. [54] Cross-sectional study of the “ICE study”	China	To investigate whether DECT, a novel quantitative technique, can be used for quantitative severity assessment in connective tissue disease-associated interstitial lung disease	147	Revolution CT, GE Healthcare, Milwaukee WI, USA No C.I.	DECT could be applied to evaluate the severity of connective tissue disease-associated interstitial lung disease.
Scharm et al. [55] Retrospective study	Germany, The Netherlands	To evaluate whether regional ventilation, lung perfusion, and late enhancement can serve as early imaging markers for disease progression in patients with idiopathic pulmonary fibrosis	32	Somatom Force^®^, Siemens Healthineers No C.I.	CT-derived functional parameters of regional ventilation and parenchymal late enhancement are potential early imaging markers for idiopathic pulmonary fibrosis progression.
Wang et al. [56] Retrospective study	China	To objectively and subjectively assess and compare the characteristics of mono-energetic images and poly-energetic images acquired by DECT of patients with breast cancer	42	SOMATOM Drive, Siemens Healthineers No C.I.	Reconstructions at low keV in the venous phase acquired by DECT improved the objective and subjective assessment of lesion conspicuity in patients with malignant breast lesions.
Volterrani et al. [57] Retrospective study	Italy	To demonstrate the feasibility of DECT for locoregional staging of breast cancer and differentiation of tumor histotypes	31	Discovery CT 750 HD, GE Healthcare No C.I.	DECT is feasible and seems to be a reliable tool for locoregional staging of breast cancer.
Moon et al. [58] Retrospective study	South Korea	To evaluate the predictive value of VMI by assessing tumor conspicuity on dual-layer spectral detector CT and correlate tumor conspicuity on VMI with prognostic biomarkers in patients with breast cancer	64	IQon Spectral CT, Philips Health System No C.I.	VMI40_DEL_ may be useful in the diagnosis of breast cancers due to higher tumor conspicuity and better enhancement than VMI40_ART_. VMI40_ART_ may be beneficial for the prediction of poor breast cancer prognoses.
Lenga et al. [59] Retrospective study	Germany	To evaluate whether breast cancer spread can be predicted by radiomic features derived from iodine maps, an application on a new generation of CT scanners visualizing tissue blood flow	77	Somatom Force, Siemens Healthineers, Forchheim, Germany. No C.I.	DECT iodine map-derived radiomic signatures have the potential to predict metastatic status in breast cancer patients. In addition, microstructural differences between primary and metastatic breast cancer tissue are also reflected by differences in the respective DECT radiomic features.
Zhang et al. [60] Prospective study	China	To evaluate the diagnostic performance of quantitative parameters derived from DECT for the preoperative diagnosis of metastatic sentinel lymph nodes, in participants with breast cancer	193	Discovery CT 750 HD, GE Healthcare No C.I.	The accuracy of the venous phase slope HU for detecting metastatic sentinel lymph nodes was 90.5% on a per-lymph node basis and 87.0% on a per-patient basis. The accuracy and specificity at venous phase slope HU were higher than their counterparts in the morphologic parameters (*p* < 0.001).
Terada et al. [61] Retrospective study	Japan	To evaluate the similarity of quantitative DECT parameters between the primary breast cancer lesion and axillary LN for predicting LN metastasis.	137	Revolution CT; GE Healthcare, Chicago, IL, USA No C.I.	The quantitative DECT parameters, including the slope HU, IC, and attenuation values at 40 keV and 70 keV, were useful for predicting LN metastasis, as previously reported. However, these DECT parameters may be influenced by differences in the CT scanner, scanning protocols, and injection protocols of the contrast medium.
Nagano et al. [62] Retrospective study	Japan	To assess the utility of ED from DECT in diagnosing metastatic mediastinal lymph nodes in patients with non-small-cell lung cancer in comparison with conventional CT and FDG PET/CT	57	IQon Spectral CT, Philips Healthcare No C.I.	ED may complement conventional CT findings and FDG uptake on PET/CT in diagnosing metastatic nodes.
Chang et al. [63] Prospective study	Korea	To investigate the diagnostic value of DECT in differentiating between low- and high-risk thymomas and thymic carcinomas.	37	Discovery CT750 HD; GE Healthcare, Wauwatosa, WI, USA, No C.I.	DECT using a quantitative analytical method based on IC measurement can be used to differentiate among thymic epithelial tumors using single-phase scanning. IHU and IC were lower in high-risk thymomas/carcinomas than in low-risk thymomas.
Zhou et al. [64] Retrospective study	China	To explore the utility of DECT parameters in distinguishing thymic epithelial tumours from thymic cysts among lesions <5 cm in diameter.	56	Discovery 750HD CT system (GE Healthcare, Madison, WI, USA No C.I.	DECT could distinguish thymic epithelial tumours from thymic cysts (d. < 5 cm). The CT value under 60 keV in the arterial phase has better diagnostic performance.
Cheng et al. [65] Retrospective study	China	To objectively and subjectively assess optimal VMI characteristics from DECT and the diagnostic performance for the T-staging in patients with thoracic esophageal cancer	68	SOMATOM Drive, Siemens Healthineers No C.I.	DECT has great advantages in evaluating T-staging in patients with EC. The venous phase VMI40 keV can improve the accuracy of evaluating T- staging, and quantitative parameters derived from DECT also can help to identify T1-2 from T3-4.
Zopfs et al. [66] Retrospective study	Germany	To investigate the diagnostic value of spectral detector DECT-derived low-keV VMI and iodine overlays for locoregional, pretherapeutic assessment of esophageal cancer	74	IQon, Philips Healthcare, Best, The Netherlands) No C.I.	Virtual monoenergetic images at 40–60 keV improve qualitative assessment of the esophageal cancer lesion and depiction of lymph nodes and vessels at pretherapeutic.
Si-Mohamed et al. [67] Retrospective study	France	To assess whether VNC images derived from contrast dual-layer DECT images could replace TNC images for aortic intramural hematoma diagnosis in acute aortic syndrome imaging protocols by performing quantitative as well as qualitative phantom and clinical studies.	21	IQon, Philips Healthcare No C.I.	Dual-layer -DECT offers similar performances with VNC and TNC images for intramural hematoma diagnosis without compromise in diagnostic image quality. VNC imaging with dual-layer DECT reduces the number of acquisitions and radiation exposure in acute aortic syndrome imaging protocol.
Perez-Johnston et al. [68] Retrospective study	USA	To evaluate the utility of perfusion defects on dual-energy CT angiograms in assessing the clinical severity of pulmonary embolism	1136	Discovery CT750 HD, GE Healthcare No C.I.	The presence of a perfusion defect correlates with several parameters evaluating pulmonary embolism severity. A perfusion defect and higher perfusion defect score were associated with a lower survival.
Abdellatif et al. [69] Meta-Analysis	Canada	To investigate the accuracy of DECT in the detection of acute	7	No C.I.	DECT shows high sensitivity, specificity, and diagnostic accuracy in the detection of acute pulmonary embolism. The high positive likelihood ratio highlights the high clinical importance of DECT as a prevalence-independent, rule-in test. Studies with a larger sample size with standardized reference tests are still needed to increase the statistical power of the study and support these findings.
Monti et al. [70] Meta-Analysis	Italy	To evaluate the diagnostic performance of DECT with regard to its post-processing techniques, namely, linear blending, iodine maps, and VMI reconstructions, in diagnosing acute pulmonary embolism	17 studies	No C.I.	DECT displayed pooled sensitivity and specificity of 0.87 and 0.93 for linear blending alone, 0.89 and 0.90 for linear blending and iodine maps, and 0.90 and 0.90 for linear blending iodine maps, and VMI reconstructions. The performance of DECT for patient management is not superior to that reported in the literature for single-energy CT (0.83 sensitivity and 0.96 specificity). DECT did not yield substantial advantages in the identification of patients with acute pulmonary embolism compared to single-energy techniques.
Flors et al. [71] Retrospective study	USA	To evaluate the diagnostic performance of dual-source DECT in the detection of endoleaks after thoracic endovascular aortic repair for thoracic aortic aneurysm and to investigate if a double-phase (arterial and dual-energy late delayed phase) or a single-phase (dual-energy late delayed phase) acquisition can replace the standard triphasic protocol	48	Somatom Definition, Siemens Healthcare No C.I.	VNC and late delayed phase images reconstructed from a single DECT acquisition can replace the standard triphasic protocol in follow-up examinations after thoracic endovascular aortic repair, thereby providing a significant dose reduction.
Shuman et al. [72] Prospective study	USA	To compare DECT aortography using a 70% reduced iodine dose to single-energy CT aortography using a standard iodine dose in the same patient	21	Discovery CT750 HD; GE Healthcare, Waukesha, WI) One author received research grants from GE Healthcare No C.I.	70% reduced iodine DECT aortography may result in similar aortic attenuation, CNR, SNR, and lower although acceptable subjective image scores when compared to standard iodine single-energy aortography in the same patient.
Kosmala et al. [73] Prospective study	Germany	To determine the diagnostic performance of DECT for the detection of bone marrow infiltration in patients with multiple myeloma by using a VNCa technique	34	Somatom Force, Siemens Healthineers,, Germany. No C.I. related to this article	Visual and ROI-based analyses of dual-energy VNCa images had excellent diagnostic performance for assessing bone marrow infiltration in patients with multiple myeloma with precision comparable to that of MR imaging.
Kosmala et al. [74] Retrospective study	Germany	To evaluate whether different MRI patterns also result in different bone marrow DECT VNCa attenuation values	53	Somatom Force; Siemens Healthineers t One author (BK) is an employee of Siemens Healthcare	Bone marrow VNCa attenuation numbers of various imaging patterns in patients with plasma cell disorders differ significantly and a diffuse imaging pattern can be determined confidently using DECT, when ROIs are carefully selected on the basis of MRI findings.
Ishiwata et al. [75] Retrospective study	Japan	To examine whether water-hydroxyapatite images improve the diagnostic accuracy of bone metastasis compared with non-contrast CT alone	83	Revolution CT, GE Healthcare, Waukesha, USA No C.I.	CT with water-hydroxyapatite images reduces the need for additional radiographic imaging, potentially reducing costs and radiation exposure.
Borggrefe et al. [76] Retrospective study	Germany	To evaluate quantitative iodine density mapping as a quantitative biomarker for the separation of vertebral trabecular bone metastases from healthy-appearing trabecular bone	43	IQon Spectral Detector CT, Philips Healthcare D.M. and J.B. received honorarium from Philips for scientific lectures	Iodine density measured yielded highest sensitivity and specificity for the statistical differentiation of vertebral trabecular metastases and healthy trabecular bone using an iodine density threshold of 4.5 mg/mL
Huang et al. [77] Retrospective study	USA	To evaluate the sensitivity, tumor conspicuity, and image quality of different material decomposition images of phantoms and patients with nearly isodense bone metastases using rapid-kilovoltage-switching DECT.	6	Discovery CT750 HD scanner, GE Healthcare B.Y. is a consultant for GE Healthcare	Dual-energy CT with hydroxyapatite–water material decomposition may improve the detection of bone marrow metastases, especially for subtle isodense tumors.

Conflict of interest (C.I.); virtual monoenergetic images (VMI); dual-energy computed tomography (DECT); lymph node (LN); electron density (ED); iodine concentration (IC); computed tomography (CT); virtual non-contrast (VNC); true non-contrast (TNC); solitary pulmonary nodules (SPNs); virtual noncalcium (VNCa); signal-to-noise ratio (SNR); contrast-to-noise ratio (CNR); region of interest (ROI); magnetic resonance imaging (MRI); Iodine-related Hounsfield unit (IHU).

## Data Availability

No new data were created or analyzed in this study. Data sharing is not applicable to this article.

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
