# Peer review of "Thoracic Diseases: Technique and Applications of Dual-Energy CT"

_diagnostics, 2023, doi:10.3390/diagnostics13142440_

Round 1

Reviewer 1 Report

Good subject of narrative review. But, reading is not easy : English language, units, definitions…

Following list is not exhaustive.

Abstract :

Last sentence is almost entirely repeated…

Introduction :

Line 48 and many others : keV instead of KeV

Table 1 :

Define DE.

“Narrative reviews were reported only if a few original articles and no systematic reviews or meta-analysis were available.” : … about anatomical/disease district ?

One table lacks : how many studies have been analyzed ? which ones ?

2.1 : Pleura

…(with optimal iodine concentration [IC] threshold was 1.3 mg/mL)… : ?

p 0.001 instead of P .001

IC or [IC] ?

Line 89 : 2.3 mg/mL ± 0.8; attenuation, AUC 0.96, 95% CI: 0.9, 0.99 : what does it mean ?

transitivity carcinoma : I cannot get this type of tumor on PubMed.

Line 114 : that underwent at non-contrast DECT imaging : a non-contrast ?

2.2 : Lung

Line 160 : cutoff value for IC was 3,10, for normalized ICC 0,29, and for slope [λ] HU 3.57 : what does it mean ? . instead of , ?

Please explain λHU.

“Authors evaluated 93 patients with lung cancer eligible for curative surgery identified in a multivariate analysis that stage and entropy are independent risk factors predicting both overall and disease-free survival” : does this sentence mean that “The authors evaluated 93 patients with lung cancer eligible for curative surgery. Multivariate analysis showed that stage and entropy were independent risk factors predicting overall survival and disease-free survival.” ?

Define FVC, HRCT and DLCO.

cm3 instead of cm3.

Line 199 : "a 60-mL contrast media" : what is the iodine concentration of the medium ?

What are “ventilation, lung perfusion and late enhancement” regarding lung DECT ?

2.3 : Breast

Line 224 : “when a patient’s breast density is greater than 25% [65]” : please explain.

Line 246 : “via a peripheral vein at a rate of 2/2.5 mL/s” : iodine concentration ?

Line 249 : “AUC 0.817 they differentiate cancer from other lesions” : ??

Line 252 : “Ki67 +” : percentage ?

Line 263 : “and accuracies of up to 92.6 in the training and 82.6” : percentages ? same remark on following values…

2.4 : Lymph nodes

Line 270 : “about 193 patients” : strange formulation.

Figure 5 : explain slope HU and iodine concentration with scatterplot.

Line 298 and following : define LN.

2.5 : Mediastinal neoplasms

Line 344 : define iodine-related HU.

Line 365 : “The IC evaluation with cut-off values of 10.99 and 11.05” : units ?

2.7 : Vascular emergency

Line 487 : “Thus, iodine quantification could be a useful tool to quantification improves the characterization of portal vein thrombosis” : ??

2.8 : Bone

Line 519 : “4.5 mg mL−1” : 4.5 mg.mL-1 or mg/mL

Discussion/synthesis is lacking.

Extensive editing of English language is required.

Author Response

All the authors thank the reviewer for his kind comments.

We proofread the manuscript to improve readability and the English language. We corrected and added all descriptions as requested. We have added comments and improved the conclusions.

Reviewer 2 Report

The review titled "Thoracic diseases: technique and applications of Dual-energy CT" is an engaging and informative narrative review that presents a comprehensive overview of the applications and benefits of Dual-energy CT (DECT) in the field of thoracic diseases. The primary focus of the review is on the latest advancements in this technology.

The authors have done a commendable job in organizing the content in a systematic manner, allowing readers to grasp the concepts and information easily. The review provides a clear understanding of the technique and principles underlying DECT, enabling readers to appreciate its relevance in the diagnosis and management of thoracic diseases.

One notable strength of the review is its emphasis on the recent innovations in DECT. By highlighting the most up-to-date developments, the authors demonstrate their awareness of the evolving landscape in the field. This aspect is particularly valuable for clinicians, researchers, and other healthcare professionals seeking to stay abreast of the latest advancements in thoracic imaging.

Moreover, the writing style employed in the review is engaging and coherent, making it accessible to a wide range of readers. The authors effectively convey complex concepts and technical details without overwhelming the audience, striking a good balance between technical accuracy and readability.

Based on the strengths mentioned above, I would recommend this review for acceptance after minor revisions. While the overall quality of the review is commendable, there may be some areas where minor improvements could further enhance the clarity and impact of the paper.

To ensure the review's excellence, I suggest the following minor revisions:

1.     Line 133-136, “In a multivariable analysis including lesion diameter, smoking history, and 133 ringlike peripheral high IC, the only independent significant predictor of pulmonary metastasis was ringlike peripheral high IC (Interobserver agreement, expressed as kappa, was 0.80). This must be citated with suitable references, ref 55 does not support the claims. Provide additional examples or case studies to further illustrate.

2.     Line 157-160, “Lesions were considered metastatic for an iodine uptake on I-131 single-photon emission CT or increased size in follow-up CT. The DECT parameters of the metastatic lesions were significantly higher than those of the benign nodules (cutoff value for IC was 3,10, for normalized ICC 0,29, and for slope [λ] HU 3.57).” The values should be rechecked, and statement should be rephrased for clarity.

3.     Conclusions: Consider expanding on the limitations or challenges associated with DECT, providing a balanced perspective on its use in clinical practice.

By addressing these suggestions, the review will be strengthened and ready for acceptance. Overall, this narrative review makes a valuable contribution to the field of thoracic imaging, and I believe it will be well-received by the scientific community.

Author Response

All the authors thank the reviewer for his kind comments.

We proofread the manuscript to include the following modifications:

1. the sentence  “In a multivariable analysis including lesion diameter, smoking history, and 133 ringlike peripheral high IC..."  is supported by ref. 56

2. The values have been rechecked.

3. Conclusion has been expanded.

Reviewer 3 Report

Authors present a narrative review of use of dual-energy computed tomography (DECT) of chest and its applications in diagnostics of thoracic diseases. The manuscript is intended to provide short state-of-the art update on daily use of the method in clinical practice. There are several clinical examples for various thoracic diseases. Drawback is that there is no explanation of criteria of choice of literature for the review. A comparison of DECT to FDG/PET CT and Photon-Counting Detector CT. I suggest to include a table on pros/cons, and include MRI and sonography as further diagnostic methods.

For Discussion I suggest to include and comment:

 Nagano H, Takumi K, Nakajo M, Fukukura Y, Kumagae Y, Jinguji M, Tani A, Yoshiura T. Dual-Energy CT-Derived Electron Density for Diagnosing Metastatic Mediastinal Lymph Nodes in Non-Small Cell Lung Cancer: Comparison With Conventional CT and FDG PET/CT Findings. AJR Am J Roentgenol. 2022 Jan;218(1):66-74. doi: 10.2214/AJR.21.26208. Epub 2021 Jul 28. Erratum in: AJR Am J Roentgenol. 2022 Feb;218(2):391. PMID: 34319164.

  Hagen F, Walder L, Fritz J, Gutjahr R, Schmidt B, Faby S, Bamberg F, Schoenberg S, Nikolaou K, Horger M. Image Quality and Radiation Dose of Contrast-Enhanced Chest-CT Acquired on a Clinical Photon-Counting Detector CT vs. Second-Generation Dual-Source CT in an Oncologic Cohort: Preliminary Results. Tomography. 2022 Jun 3;8(3):1466-1476. doi: 10.3390/tomography8030119. PMID: 35736867; PMCID: PMC9227736.  

Acceptable. 

Author Response

All the authors thank the reviewer for his kind comments.

We proofread the manuscript to improve readability and decided to include the suggested articles. An explanation of the criteria of choice of literature for the review is in table 1.

Round 2

Reviewer 1 Report

The manuscript is now well readible.

Still persists : 

- all along : please write keV instead of KeV ;

- all along : within decimal numbers, use a . instead of a , (the English language requires this rule).

English is now much better, as are scientific notations.

Author Response

Thanks to the reviewer for the suggestions. The suggested changes will be implemented.

Reviewer 3 Report

The authors have sufficiently responded to reviewer remarks. 

Acceptable. 

Author Response

Thanks to the reviewer for the suggestions.